# SPARDACUS SafetyCage:
# A new misclassification detector

Pål Vegard Johnsen[*1], Filippo Remonato[1], Shawn Benedict[2], and Albert Kwesi Ndur-Osei[2]

[1]SINTEF Digital, Oslo, Norway
[2]Department of Electrical and Computer Engineering, University of Waterloo, Waterloo, ON, Canada
[*]pal.johnsen@sintef.no

## Abstract

Given the increasing adoption of machine learning techniques in society and industry, it is important to put procedures in place that can infer and signal whether the prediction of an ML model may be unreliable. This is not only relevant for ML specialists, but also for laypersons who may be end-users. In this work, we present a new method for flagging possible misclassifications from a feed-forward neural network in a general multi-class problem, called SPARDA-enabled Classification Uncertainty Scorer (SPARDACUS). For each class and layer, the probability distribution functions of the activations for both correctly and wrongly classified samples are recorded. Using a Sparse Difference Analysis (SPARDA) approach, an optimal projection along the direction maximizing the Wasserstein distance enables $p$-value computations to confirm or reject the class prediction. Importantly, while most existing methods act on the output layer only, our method can in addition be applied on the hidden layers in the neural network, thus being useful in applications, such as feature extraction, that necessarily exploit the intermediate (hidden) layers. We test our method on both a well-performing and underperforming classifier, on different datasets, and compare with other previously published approaches. Notably, while achieving performance on par with two state-of-the-art-level methods, we significantly extend in flexibility and applicability. We further find, for the models and datasets chosen, that the output layer is indeed the most valuable for misclassification detection, and adding information from previous layers does not necessarily improve performance in such cases.

## 1 Introduction

A crucial consideration when deploying a machine learning (ML) model in real-life applications is the ability to infer how reliable the predictions are. As an example, consider a model used to detect a hazardous situation within an industrial facility. First, it is important that the model can capture an unsafe situation as it happens to then signal the operators. Second, to achieve trust, the model should not raise false alarms too frequently, or future warnings lose credibility. Thus, to deploy the ML model in a real-world setting, the reliability of its predictions need to be considered in some way prior to making actual decisions (such as stopping the production line).

In most neural network (NN) classifiers, a softmax activation function is utilized on the output layer to interpret each value as the probability of belonging to a particular class. The class prediction for any sample is most often equal to the class with the largest softmax probability in the output layer.

It has been shown that misclassifications may arise even when the largest softmax probability is close to one [1]. Nonetheless, a pattern was discovered where the maximum softmax probability tended to be smaller for incorrectly classified samples than for correctly classified samples. This discovery was used to make a simple threshold-based misclassification detector, see [2], called Maximum Softmax Probability (MSP) Detector.

In [3], a method named DOCTOR for misclassification detection was proposed, based on an approximation of the misclassification probability, $\mathrm{Pe}(x)$, for a particular sample $x$ by only using the softmax output layer values, $P_{\hat{Y}|X}(c \mid x)$, for each class $c$ of total $C$ classes. In particular $\mathrm{Pe}(x) \approx 1 - \sum_{c=1}^{C} P_{\hat{Y}|X}^2(c \mid x)$. The method flags a prediction as untrustworthy whenever the odds of a misclassification event is larger than some threshold.

From our literature search, it is apparent that the DOCTOR and MSP-detector methods represent the current state of the art of misclassification detection, which we use for comparison to SPARDACUS.

The *SafetyCage*, introduced in [4], is another misclassification detector. This statistical framework collects the pre-activation vector in each layer, and assumes the corresponding multivariate probability density function (PDF) of correctly predicted in-distribution samples to follow a Gaussian distribution. These PDFs, per class, are fitted to the training data of the NN model. To infer the uncertainty of a class prediction, the Mahalanobis distance, inspired by the approach described in [5], is used to measure the likeliness that the pre-activation values, in

---

[*]Corresponding Author.

Proceedings of the 6th Northern Lights Deep Learning Conference (NLDL), PMLR 265, 2025.

each layer, are generated from the fitted Gaussian distribution of correctly predicted samples. This Mahalanobis-based SafetyCage was tested on two feed-forward neural networks trained on benchmark datasets MNIST and CIFAR-10, respectively. The classifier trained on the MNIST dataset had an accuracy of 0.93, whereas the one trained on CIFAR-10 had an accuracy of 0.48. It was observed that for the well-performing MNIST model, the multivariate Mahalanobis SafetyCage was able to detect and flag 60% of the wrong classifications. On the other hand, for the CIFAR-10 model with poor performance, the SafetyCage was no better than random guessing. After closer inspection, the assumption of Gaussianity of the pre-activation vector for the CIFAR-10 model was not accurate [4].

While the Mahalanobis-based SafetyCage only uses samples that are correctly predicted, we propose the SPARDACUS method for misclassification detection that uses both correctly and wrongly classified samples. We compare the results of SPARDACUS to the previous SafetyCage, the DOCTOR method, and the MSP-detector method.

We note that a task related to the detection of misclassifications is what is called *out-of-distribution* (OOD) detection, where the aim is to detect whenever an input sample is inherently different from the data used during training of the model, and hence the corresponding prediction should not be trusted. However, the most insidious misclassifications happen with in-distribution data, for which the model would be assumed to work correctly, and not OOD-data. Indeed, in [6] it is shown that the best OOD-detector is not always the best at detecting NN classification errors. The authors in [6] further emphasize that if the focus is on use of NNs in safety-critical applications, misclassification detection should be the paramount focus, and not OOD-detection. For these reasons, this work focuses on misclassification detection using in-distribution data, and will not draw a comparison to OOD-detection methods.

## 2 Methods

### 2.1 SPARDACUS

Consider the function $\mathcal{F}_{i,l}$ which corresponds to the PDF that generates the activation values at layer $l$ for a sample correctly predicted as class $i$ by the NN classifier. Conversely, let $\mathcal{G}_{i,l}$ correspond to the PDF that generates the activation values for incorrectly classified samples. Given these two PDFs, one may infer which distribution a new test sample $x'$ belongs to by designing a decision procedure to flag wrongly classified samples. In principle this is a binary classification problem to which we could apply any ML method to *predict* if $x'$ is correctly

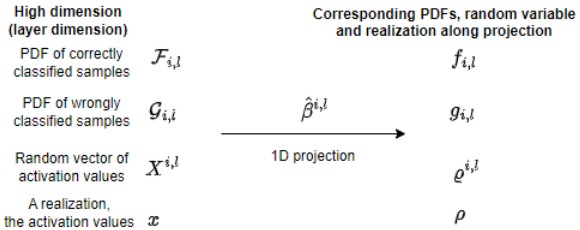

**Figure 1.** Notation in high dimension and corresponding notation in dimension 1 after the random vector $X^{i,l}$ is projected onto the one-dimensional subspace defined by $\hat{\beta}^{i,l}$ for class $i$ and layer $l$.

or wrongly classified; but if $\mathcal{F}_{i,l}$ and $\mathcal{G}_{i,l}$ could be approximated directly, statistical tests backed by solid theory would become available. A direct application of this procedure is however challenging, since the dimensionality of the PDFs is linked to the size of the NN classifier's layers; i.e. large layers imply high-dimensional PDFs. To combat this, we project the data for each layer and class to one dimension, effectively collapsing the multi-dimensional PDFs into one-dimensional ones. The goal is to obtain two PDFs along this 1D-projection, denoted $f_{i,l}$ and $g_{i,l}$, which are minimally overlapping. The PDFs are estimated using the same training data the classifier is constructed from.

To this end, Mueller et al. [7] propose an approach referred to as a Sparse Differences Analysis (SPARDA), which given observed samples from two multivariate PDFs, searches for the optimal projection maximising the Wasserstein distance between the projected 1D empirical distribution functions (ECDFs). This is a non-smooth, non-concave optimization problem. We apply the fastSPARDA optimization algorithm available at https://bitbucket.org/jwmueller/principal-differences-analysis/src/master/. The optimization problem includes a regularization parameter $\lambda$ to induce sparsity in the projection. We denote the projection direction given by SPARDA as $\hat{\beta}^{i,l}$, and $\varrho^{i,l}$ as the projected random variable of activations along $\hat{\beta}^{i,l}$. Figure 1 summarises the projection operation and notation.

If a new sample $x'$ is predicted to be a member of class $i$, one can infer at any given layer $l$ whether it is more likely generated from $f_{i,l}(\rho)$ or $g_{i,l}(\rho)$ by inspecting the observed value $\rho$ along the projection. However, $f_{i,l}(\rho)$ and $g_{i,l}(\rho)$ are unknown. To overcome this, we fit each PDF as a Gaussian mixture model (GMM) due to its flexibility and computational efficiency. A typical situation with overlapping can be seen in Figure 2 showing histograms of $f_{7,-1}$ and $g_{7,-1}$ on training data and test data for the class (digit) 7 from MNIST, with $l = -1$ indicating the output layer.

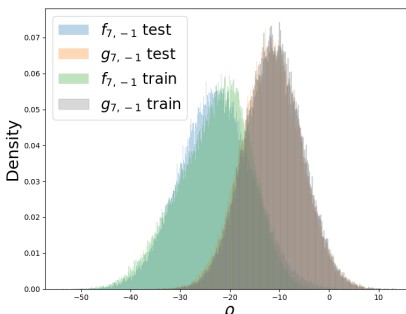

**Figure 2.** Illustration of the PDFs for $f_{7,-1}$ and $g_{7,-1}$ on both the training data and the test data. This represents the digit 7, output layer, and the MNIST model.

## 2.2 Inference

With inspiration from the likelihood ratio test we define the statistic, $S^{i,l}(\varrho^{i,l})$, using the two aforementioned PDFs for a random variable $\varrho^{i,l}$ with observed values along the projection $\hat{\beta}_{i,l}$:

$$S^{i,l}(\varrho^{i,l}) = -\ln\left(\frac{f_{i,l}\left(\varrho^{i,l}\right)}{g_{i,l}\left(\varrho^{i,l}\right)}\right). \qquad (1)$$

Moreover, consider the statistic $S_C^{i,l}(\varrho_f^{i,l})$ where $\varrho_f^{i,l} \sim f_{i,l}(\rho)$. The corresponding PDF of $S_C^{i,l}$, denoted $h_C(s_c^{i,l})$, is the PDF we expect for samples with correct class prediction $i$. The larger the value of $s^{i,l}$, the more unlikely it is generated from $h_C(s_C^{i,l})$. We define the following hypothesis test for a test sample $x'$ with class prediction $\hat{y}'$:

$$H_0\colon S^{\hat{y}',l} \sim h_C(s^{\hat{y}',l}) \quad \text{vs.} \quad H_1\colon S^{\hat{y}',l} \not\sim h_C(s^{\hat{y}',l}). \qquad (2)$$

A corresponding $p$-value for an observed value $s^{\hat{y}',l}$ can be computed as $p_{S_C} = P(S_C^{\hat{y}',l} \geq s^{\hat{y}',l}) = 1 - H_C(s^{\hat{y}',l})$ with $H_C(s^{\hat{y}',l})$ the CDF of $S_C^{\hat{y}',l}$. Recall that a $p$-value under the null hypothesis will follow a uniform probability distribution ($p_{S_C}^{H_0} \sim U(0,1)$). A small $p$-value indicates the unlikeliness that $s^{\hat{y}',l}$ is generated from $f_{\hat{y}',l}(\rho)$. Rather, it is an indication that $s^{\hat{y}',l}$ is generated from $g_{\hat{y}',l}(\rho)$. The null hypothesis can then be rejected, ultimately flagging a prediction as wrong, for a $p$-value less than a predefined significance level $\alpha_C$.

Notice that one can also define the statistic $S_W^{i,l}(\varrho_g^{i,l})$ where $\varrho_g^{i,l} \sim g_{i,l}(\rho)$. Using this statistic instead, one can construct in the same way a hypothesis test with significance level $\alpha_W$. See Appendix A for more information.

We emphasize that, unlike the Mahalanobis SafetyCage [4], the SPARDACUS SafetyCage does not rely on pre-activation values, as there is no longer need to assume Gaussianity.

### 2.2.1 Estimation of $p$-values from the S statistics

As the PDFs $f_{i,l}$ and $g_{i,l}$ are estimated as a Gaussian mixture model, the corresponding PDFs of $S_C^{i,l}$ and $S_W^{i,l}$ are not always easily accessible. An easy way to estimate the PDFs of $S_C^{i,l}$ and $S_W^{i,l}$ is via Monte Carlo simulation. After a specified number of repeated generations from $f_{i,l}$ and $g_{i,l}$, we can compute the ECDFs of $S_C^{i,l}$ and $S_W^{i,l}$, denoted $\hat{F}_C^{i,l}(s_{i,l})$ and $\hat{F}_W^{i,l}(s_{i,l})$. From this we can estimate the corresponding $p$-values as:

$$\hat{p}_{S_C}^l(s_{\hat{y}',l}) = 1 - \hat{F}_c^{\hat{y}',l}(s_{\hat{y}',l}),$$

and

$$\hat{p}_{S_W}^l(s_{\hat{y}',l}) = \hat{F}_w^{\hat{y}',l}(s_{\hat{y}',l}).$$

Whether to use $S_C^{i,l}$ or $S_W^{i,l}$ to get the most robust results will be based on the accuracies of the estimated PDFs $f_{i,l}$ and $g_{i,l}$.

The notation so far has been focused on a particular layer $l$. By our method, each layer $l$ provides a $p$-value. These $p$-values can be combined into one, using any $p$-value combination test. To this end, we will apply the Cauchy combination test as it is robust for statistically correlated $p$-values [8]. We denote $\hat{p}_{S_C}$ and $\hat{p}_{S_W}$ the final $p$-values when using either statistic $S_C$ or $S_W$, respectively. Algorithms for the method is given in B.1 and B.2 in the Appendix where we separate between the training phase of the SPARDACUS method, and the subsequent detection procedure.

## 3 Results

We will evaluate our method by using the same setup as in [4], where a feed-forward neural network is trained on MNIST, yielding a well-performing model (accuracy of 0.98), and on CIFAR-10 yielding a poor-performing model (accuracy of 0.48). The neural network consists of two hidden layers, with 256 and 128 neurons respectively, and ReLu activation functions, along with an output layer featuring ten neurons with Softmax activation. Training utilized the standard Adam optimizer. Once the model is trained, our method estimates the projections $\hat{\beta}_{i,l}$ and corresponding PDFs $f_{i,l}, g_{i,l}$ for all classes and layers (except the input layer, i.e. the images). On a held-out test data disjoint from the data the NN model was trained on, we evaluate our misclassification detector for different values of $\alpha_C$ and $\alpha_W$. Additionally, we present tables showing the precision, recall, specificity, negative predictive value as well as the MCC for the best performing $\alpha$ value.

We advocate for using the Matthews correlation coefficient (MCC) as the most meaningful performance metric to evaluate any binary classification model [9]. The MCC ranges from -1, meaning the

classifier is always wrong, to 1 meaning a perfect classifier. A coin tossing classifier with a 50 % chance to assign a prediction to either of the two classes will give MCC = 0 [9]. Also by definition, a classifier that predicts only one class every time will give MCC = 0 (see [9] for more details). We compare our method with the MSP-detector method introduced in [2], the DOCTOR method [3], and the SafetyCage method introduced in [4]. The threshold in SPARDACUS and in [4] can be interpreted as the same, as they both are based on $p$-values. The threshold for the MSP-detector method is with respect to the probability of the prediction, while for the DOCTOR method it is with respect to the estimated odds of misclassification.

For the Mahalanbois SafetyCage and the SPARDACUS method, we evaluate the results for different layer aggregations (combining $p$-values from different layers) to investigate the information that is covered in the different layers. Specifically, we investigate when only applying the output layer (out), the penultimate layer (pen) or all hidden layers plus the output layer (all).

## 3.1 SPARDACUS Results

In all results to come, the SPARDA projection $\hat{\beta}_{i,l}$ was computed by setting $\lambda = 0$, imposing no regularization, and using the fastSPARDA algorithm. We used Monte Carlo simulations to estimate the CDFs of the $S$ statistics by generating 1 million samples.

Tables 1 and 2 show the best results, ranked with respect to the MCC, for the four methods on the MNIST model and CIFAR-10 model respectively when the optimal threshold as well as the parameters and PDFs for the Mahalanobis SafetyCage and SPARDACUS methods are estimated on the training data. The methods are evaluated on the unseen test data.

Notice that for all methods, only using the output layer turned out to give the largest MCC value. The same applies to the SafetyCage Mahalanobis, an aspect that was not investigated in the original paper [4] where only hidden layers up to the penultimate layer were investigated.

We also show the precision, recall, specificity, and negative predictive value (NPV) for each case. While the outputs of the MSP and DOCTOR methods are deterministic, the 1D-projections calculated in SPARDACUS using the SPARDA algorithm, result in stochasticity. For this method we therefore show the result of five runs in terms of mean and standard deviation. Notice the small variation in MCC. After closer inspection, this is due to nearly equal 1D-projection.

In Figures 3 and 4, for each of the misclassification detectors, we show how the MCC varies for different threshold values on the test data for MNIST

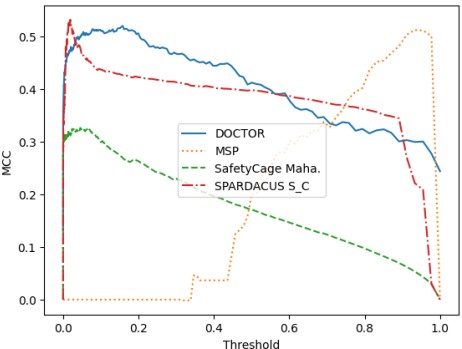

**Figure 3.** Plots showing thresholds vs MCC for the misclassification detectors for the test data with the MNIST model.

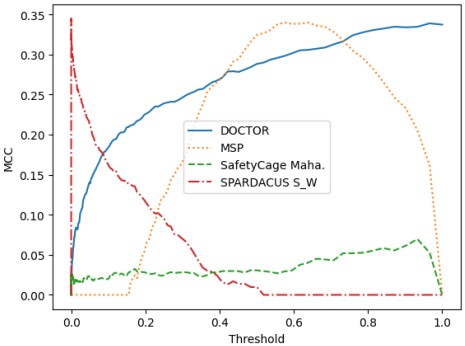

**Figure 4.** Plots showing thresholds vs MCC for the misclassification detectors for the test data with the CIFAR-10 model.

and CIFAR-10 respectively. We see that the SPARDACUS method is most sensitive to the choice of threshold with respect to performance, with a sharp peak for the optimal threshold. At the same time, if we compare the MCC values at the peaks for each method, the SPARDACUS method achieves by a small margin the highest MCC values for both datasets. See Table C.1 in Appendix C where the MCC values at the peaks for the SPARDACUS method are extracted. Here, we also include results when applying different sets of layers for the SPARDACUS method which confirms that only using the output layer yields the highest MCC values.

In practice we do not know in advance what the optimal threshold is. To estimate the threshold on the same data that the ML model is trained on can make sense in terms of utilizing all data available, particularly for data-driven methods such as the Mahalanobis SafetyCage and the SPARDACUS, where also PDFs are fitted to data. However, in certain situations the training data is not available, and instead new data must be collected during deployment. Moreover, using the training data may lead to overfitting and less generalization capabilities. For comparison, we include the scenario where the parameters needed for each misclassification detection

| Method | $S$ | L | Threshold | Prec. | Recall | Spec. | NPV | MCC |
|--------|-----|---|-----------|-------|--------|-------|-----|-----|
| DOCTOR | — | out | 1.13E-01 | 0.409 | 0.662 | 0.978 | 0.992 | 0.507 |
| MSP | — | out | 9.46E-01 | 0.408 | 0.658 | 0.978 | 0.992 | 0.504 |
| SPARDACUS | $S_C$ | out | 9.41E-03 | 0.457 | 0.554 | 0.985 | 0.990 | $0.491 \pm 0.010$ |
| SafetyCage Maha. | — | out | 8.98E-03 | 0.213 | 0.518 | 0.957 | 0.988 | 0.310 |

**Table 1.** Results ordered by the MCC score for each method on the MNIST dataset when the threshold is optimized on the training data, and evaluated on the test data. Due to Stochasticity in the SPARDACUS method we do five runs, and present the performance metrics with the respect to the average, and additionally the MCC with $\pm$ standard deviation. The SafetyCage Mahalanobis method is taken from [4]. The MSP-detector and DOCTOR methods utilise a threshold $T$, while SPARDACUS and SafetyCage have a significance level $\alpha$.

| Method | $S$ | L | Threshold | Prec. | Recall | Spec. | NPV | MCC |
|--------|-----|---|-----------|-------|--------|-------|-----|-----|
| SPARDACUS | $S_W$ | out | 3.67E-06 | 0.646 | 0.783 | 0.550 | 0.707 | $0.343 \pm 0.003$ |
| MSP | — | out | 6.27E-01 | 0.636 | 0.807 | 0.515 | 0.718 | 0.338 |
| DOCTOR | — | out | 9.99E-01 | 0.623 | 0.858 | 0.462 | 0.744 | 0.337 |
| SafetyCage Maha. | — | out | 8.98E-03 | 0.213 | 0.51 | 0.15 | 0.57 | 0.070 |

**Table 2.** Same procedure and results as for Table 1, however with respect to the CIFAR-10 dataset.

method, including the threshold value, are estimated based on data never used by the ML model. We do this by randomly splitting the test data (10 000 samples) equally in two subsets, estimate the parameters on the first subset, and evaluate the detection methods on the second subset. To account for difference in performance with respect to the splitting of data, we repeat the process five times each with random splitting of the test data. The results are given in Tables 3 and 4.

Based on the experiments and following results we see that the SPARDACUS method is superior to the SafetyCage Mahalanobis method from [4]. Moreover, compared to the MSP-detector and DOCTOR methods, the SPARDACUS method using the $S_C$ and $S_W$ is essentially on par.

As expected, all methods performed much better on the well-performing model, due to there being more useful information encoded in the activation values that could be extracted and used to flag incorrect predictions. Interestingly, the output layer was by far the most useful layer, which aids to show how this is a markedly different problem compared to OOD where the output layer can be less informative [5]. An interesting observation can be made regarding the SPARDACUS method yielding the best results when evaluating only the output layer. When inspecting the fitted projection at the output layer, $\hat{\beta}_{i,-1}$, and specifically looking at using $S_C$ for the MNIST model, we see that for every predicted class, the associated projection vector has its maximal-value element in the position corresponding to the predicted class itself. In fact, on average the maximum value along the class prediction dimension was 2.88 times larger than the second largest element. This shows that the projection vector for a specific class is heavily dominated by the dimension along the class itself. This is in fact in correspon-

dence with the MSP-detector method, where the corresponding projection would have zero-elements along all dimensions except for the class dimension. This shows that, in the special case of being applied to only the output layer, SPARDACUS can be seen as an *extension* of the MSP-detector method.

In all cases, $S_C$ performed better for the well-performing model, while $S_W$ was better for the poor-performing model.

## 4 Discussion and Conclusion

In this work, we presented a method to infer whether a particular sample is wrongly classified by an underlying NN model. SPARDACUS is based on a SPARDA projection maximizing the Wasserstein distance of the PDFs of the samples that were correctly and wrongly classified, and a hypothesis test inspired by the likelihood-ratio test. SPARDACUS can be applied at any stage of an NN classifier, and with an easy extension it could also draw information from any arbitrary combination of layers.

We tested SPARDACUS on two simple NN classifiers, one well-performing trained on MNIST and one poorly performing trained on CIFAR-10. The results have further been compared with three pre-existing methods from the literature: The DOCTOR method, the Mahalanobis-based method presented in [4], and the MSP-detector method [2].

Our results show that SPARDACUS significantly outperforms the Mahalanobis SafetyCage [4], where in particular, detection performance on the inaccurate classifier for CIFAR-10 has been improved by an order of magnitude. The MSP, DOCTOR and SPARDACUS are essentially performing at comparable levels with respect to the MCC values as shown in Tables 1, 2, 3 and 4. It is worth not-

| Method | $S$ | L | Threshold | Prec. | Recall | Spec. | NPV | MCC |
|---|---|---|---|---|---|---|---|---|
| MSP | — | out | 9.33E-01 | 0.434 | 0.629 | 0.980 | 0.991 | $0.509 \pm 0.0159$ |
| DOCTOR | — | out | 1.24E-01 | 0.415 | 0.657 | 0.979 | 0.992 | $0.507 \pm 0.023$ |
| SPARDACUS | $S_C$ | out | 3.19E-02 | 0.373 | 0.700 | 0.974 | 0.994 | $0.496 \pm 0.023$ |
| SafetyCage Maha. | — | out | 2.26E-02 | 0.173 | 0.613 | 0.934 | 0.991 | $0.295 \pm 0.027$ |

**Table 3.** Results ordered by the MCC score for each misclassification detetction method on the MNIST dataset when the threshold is optimized on half the test data (5000 samples), and the methods are evaluated on the other half. The threshold and performance metrics are presented with the average value based on five random splits of the test data. The variation in MCC is additionally presented in terms of $\pm$ standard deviation.

| Method | $S$ | L | Threshold | Prec. | Recall | Spec. | NPV | MCC |
|---|---|---|---|---|---|---|---|---|
| DOCTOR | — | out | 9.55E-01 | 0.618 | 0.862 | 0.443 | 0.755 | $0.338 \pm 0.012$ |
| MSP | — | out | 5.98E-01 | 0.645 | 0.772 | 0.554 | 0.701 | $0.336 \pm 0.009$ |
| SPARDACUS | $S_W$ | out | 3.20E-06 | 0.649 | 0.749 | 0.579 | 0.690 | $0.333 \pm 0.013$ |
| SafetyCage Maha. | — | out | 6.53E-01 | 0.524 | 0.634 | 0.401 | 0.528 | $0.043 \pm 0.015$ |

**Table 4.** Same procedure and results as for Table 3, however with respect to the CIFAR-10 dataset.

ing how SPARDACUS is an extension of the MSP-detector, being in principle (for large $\lambda$) equivalent to it when only considering the output layer. However, in contrast to the MSP and DOCTOR method, SPARDACUS can draw information from not only the output layer, but also hidden layers in the NN classifier. For the particular examples of datasets/-models shown in this work, including information from the hidden layers has not shown a definite advantage when compared to only using the output layer's information. Nonetheless, we regard it as an interesting research path to investigate whether this is always the case, or whether modifications are needed to take more advantage of the information in the hidden layers. Of all investigated methods, the performance of the SPARDACUS method is most sensitive to the choice of threshold. Clearly, SPARDACUS is more involved than the MSP-detector and DOCTOR methods both in terms of theoretical background and computational complexity. Even though MSP, DOCTOR and SPARDACUS are close in performance with respect to the MCC, we regard SPARDACUS to have three advantages: Firstly, SPARDACUS is most flexible. The flexibility that the three methods share is the choice of threshold. Other than that the MSP and DOCTOR method is static in terms of deterministic computations (in the output layer only), while the SPARDACUS method can be investigated further in several ways. Here, we list some aspects to investigate: 1) We have used the Wasserstein distance, but one may substitute it with other statistically-relevant distances. 2) The SPARDA projection can be investigated further using other optimization algorithms. 3) The importance from different layers can be weighted during training, e.g. by deploying Cauchy weights in the Cauchy combination test. 4) Experimenting with the $\lambda$ regularization parameter, or other significant statistics in place of $S^{i,l}$ is possible. Sec-

ondly, SPARDACUS can be refitted and updated for newly labelled data of misclassifications, in terms of projections and PDFs, while the MSP-detector and DOCTOR methods can only be updated with respect to the threshold. Thirdly, by being able to inspect not only the output layer, but also the inner working of the neural network in terms of previous layers, we believe that the framework SPARDACUS builds on can help us not only answer the question *whether* a classification is false, but also *why* it is false.

# Acknowledgments

This research has been funded by the The Research Council of Norway, grant 304843 (EXAIGON) as well as by the EU HORIZON-IA program, grant number 101121042 (THEMIS).

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

## A The statistic $S_W$

In a similar fashion as for the statistic $S_C$, define the statistic

$$S_W^{\hat{y}',l} = -\ln\left(\frac{f_{\hat{y}',l}\left(\varrho_g^{\hat{y}',l}\right)}{g_{\hat{y}',l}\left(\varrho_g^{\hat{y}',l}\right)}\right), \qquad (3)$$

where $\varrho_g^{i,l} \sim g_{i,l}(\rho)$. Using this statistic, one can construct in the same way as for the statistic $S_C$, develop a hypothesis test:

$$H_0: S^{\hat{y}',l} \sim h_W(s^{\hat{y}',l}) \quad \text{vs.} \quad H_1: S^{\hat{y}',l} \not\sim h_W(s^{\hat{y}',l}). \qquad (4)$$

The null hypothesis in this case shows that the sample $x'$ is wrongly classified, and the $p$-value is computed as $p_{S_W} = P(S_w^{\hat{y}',l} \leq s^{\hat{y}',l})$, this time as a right-sided test since a small observed $s^{\hat{y}',l}$ now indicates the sample $x'$ is correctly classified. The null hypothesis is rejected for a predefined significance level $\alpha_w$.

## B Algorithms

---

**Algorithm B.1** Training phase of SPARDACUS

---

Consider a dataset $D = \{x_k, y_k, \hat{y}(x_k)\}_{k=1}^N$ of input samples $x_k$, true class labels $y_k \in \{i\}_{i=1}^C$ and model predictions $M(x_k) = \hat{y}_k$. The activation values of layer $l$ for sample $k$ in model $M$ is denoted as $a_{k,l}^{(M)}$. Define statistic $S = S_C$ or $S = S_W$. Let $Q$ be number of Monte Carlo simulations. Let $T^{i,l}$, $F^{i,l}$ include activation values in layer $l$ for correct and incorrect predictions of class $i$.

**for** $l$ in $L$ **do**
  **for** i in $1, \ldots, C$ **do**
    ▷ $P_i = \{k \mid \hat{y}(x_k) = i\}$: Extract samples predicted to belong to class $i$. $T^{i,l}, F^{i,l} \leftarrow \emptyset$
    **for** $k \in P_i$ **do**
      ▷ $x^{k,l} = a_{k,l}^{(M)}$.
      **if** $\hat{y}(x_k) = y_k$ **then**
        ▷ $T^{i,l} \leftarrow T^{i,l} \cup \{x^{k,l}\}$: Add activations to set $T^{i,l}$.
      **else**
        ▷ $F^{i,l} \leftarrow F^{i,l} \cup \{x^{k,l}\}$: Add activations to set $F^{i,l}$.
      **end if**
    ▷ $\hat{\beta}^{i,l} = SPARDA(T^{i,l}, F^{i,l}, d_w)$: Use SPARDA to compute projection $\hat{\beta}^{i,l}$ that maximizes the Wasserstein distance, $d_w$, between $T^{i,l}$ and $F^{i,l}$.
    ▷ Estimate $f_{i,l}(\rho)$ as GMM from the samples along $\hat{\beta}_{i,l}$ in $T^{i,l}$.
    ▷ Estimate $g_{i,l}(\rho)$ as GMM from the samples along $\hat{\beta}_{i,l}$ in $F^{i,l}$.
    **if** $S = S_C$ **then**
      ▷ $\rho_1, \ldots, \rho_Q \sim f_{i,l}(\rho)$: Monte-Carlo generations
      ▷ Compute $Q$ realizations from the statistic in (1).
      ▷ Estimate CDF of $S_C$ via ECDF of the $Q$ realizations and collect.
    **else if** $S = S_W$ **then**
      ▷ $\rho_1, \ldots, \rho_Q \sim g_{i,l}(\rho)$: Monte-Carlo generations
      ▷ Compute $Q$ realizations from the statistic in (1).
      ▷ Estimate CDF of $S_W$ via ECDF from the $Q$ realizations and collect.
    **end if**
  **end for**
  **end for**
**end for**

---

Algorithm B.1 outlines the training phase of the SPARDACUS method where the 1D projections are estimated followed by PDFs fitted to data along the projections for correctly and wrongly classified predictions. Algorithm B.2 outlines the deployment phase of the SPARDACUS method for a new incoming data point $\{x', \hat{y}'\}$.

---

**Algorithm B.2** Inference phase of SPARDACUS

---

Given input $x'$ with model prediction $\hat{y}'$, and threshold $\alpha \in [0, 1]$.

**for** $l$ in $L$ **do**

   ▷ Compute $p$-value based on estimated CDF (ECDF) of $S^{\hat{y}',l}$.

**end for**

▷ Compute Cauchy combination test statistic $p_{cauchy}$ from observed $p$-values from investigated layers [8].

▷ Declare prediction $\hat{y}'$ as false if $p_{cauchy} < \alpha$

---

# C   All SPARDACUS results

Table C.1 shows the maximum MCC values achievable for SPARDACUS method on the test data for several different parameter sets including which layers are used (output, hidden, penultimate, all) and which statistic ($S_C$ or $S_W$).

| Data | Method | S | L | Thresh. | Prec. | Recall | Spec. | NPV | MCC |
|------|--------|---|---|---------|-------|--------|-------|-----|-----|
| MNIST | SPARDACUS | $S_C$ | out | 1.46E-02 | 0.420 | 0.700 | 0.970 | 0.993 | $0.529 \pm 0.002$ |
| MNIST | SPARDACUS | $S_C$ | all | 3.25E-02 | 0.332 | 0.721 | 0.967 | 0.993 | $0.473 \pm 0.001$ |
| MNIST | SPARDACUS | $S_W$ | out | 2.43E-06 | 0.391 | 0.551 | 0.980 | 0.990 | $0.450 \pm 0.014$ |
| MNIST | SPARDACUS | $S_C$ | pen | 4.08E-02 | 0.207 | 0.406 | 0.965 | 0.986 | $0.267 \pm 0.001$ |
| CIFAR | SPARDACUS | $S_W$ | all | 6.80E-06 | 0.645 | 0.790 | 0.543 | 0.712 | $0.345 \pm 0.001$ |
| CIFAR | SPARDACUS | $S_W$ | out | 3.28E-06 | 0.650 | 0.773 | 0.563 | 0.703 | $0.344 \pm 0.001$ |
| CIFAR | SPARDACUS | $S_C$ | out | 3.80E-01 | 0.650 | 0.690 | 0.609 | 0.652 | $0.301 \pm 0.000$ |
| CIFAR | SPARDACUS | $S_C$ | all | 5.37E-01 | 0.604 | 0.774 | 0.486 | 0.664 | $0.255 \pm 0.001$ |
| CIFAR | SPARDACUS | $S_W$ | pen | 1.24E-01 | 0.593 | 0.692 | 0.500 | 0.608 | $0.196 \pm 0.000$ |

**Table C.1.** Results for the SPARDACUS method when applying different sets of layers for both the MNIST and CIFAR-10 datasets.

