# OpenReview forum: "SPARDACUS SafetyCage: A new misclassification detector"
_NLDL.org/2025/Conference — NLDL 2025 Oral_

### Official Review · Reviewer_eSSQ · 2024-10-08
**Statistical multi-layer misclassification detection for deep neural networks extending previous work**

**Confidence:** 4

**Summary:**

Paper presents an approach to infer reliability of machine learning, and here specifically (deep) neural network classifiers by introducing misclassification detection techniques that could be used for producing more safety AI systems, and which can be seen very important topic. Previously there have been several model-agnostic and neural network -specific solutions (e.g., MSP and DOCTOR) based on output (softmax)-layer only with thresholding the maximum output or approximating the misclassification probability. Proposed approach, instead, considers combination of any layer activation values as model inputs, as in the previous SafetyCage approach. Previous work is relying only on the probability distribution of correctly classified examples, Gaussian approximation, and Mahalanobis distance -based thresholding, whereas, to increase to robustness of previous methods, the proposed approach (SPARDACUS) utilises couple of new improvements: 1) using both the PDFs of correctly and incorrectly classified activation values, 2) applying projection techniques to produce 1D PDF from original high-dimensional PDF and combine these to mixture Gaussian distribution from different layers, and 3) likelihood test -inspired statical test for final detection. Together these brings novelty to misclassification detection research. Experimental results on two image classification benchmarks show similar or on par accuracy compared to state-of-the-art methods (MSP, Doctor), and outperforming original SafetyGace method. Also, interesting result is that in both cases output-layer is the most important one for the detection.
Proposed method is sound and well-formulated. However, based on the result, the full potential of proposed approach against output-layer only methods is not indicated with these particular empirical datasets.

**Strengths:**

Paper is clearly written and structured, and pretty easy to follow. The building blocks of the proposed method are well-justified in relation to previous approach, showing an interesting techniques to combine reliability information from different network layers. Although, building blocks by themselves are not novel, the combination gives new approach to misclassification detections, improving the previous methods on empirical evaluations and performs on par with state-of-the art output-layer-based methodologies, providing some interesting results and fresh ideas.

**Weaknesses:**

Although proposed approach is interesting with some novelties, empirical evaluations are not fully convincing, showing the usefulness and full potential of multi-layer misclassification detection, at least, on these benchmark image classification datasets. Here some questions related to results section:

- Why the results in the main text (Table 1 and 2) are shown with detection threshold optimized on test sets? That is not very practical, and should be based on training or validation data instead? (some of these are presented in appendix, but are, in my opinion, more important than those in the main text currently).
- How sensitive the different methods are for choosing and optimizing the detection threshold (e.g., related to shape of the curves in Figures 3 and 4)?

**Justification:**

Paper is well-written and structured, providing a novel combination of techniques to misclassification detection domain. The results are in line with some of the state-of-the-art methodologies. Although paper gives some interesting results, full potential and benefits of using multi-layer detection are not shown; deeper analysis of the properties of the proposed methodologies and additional datasets from different application domains could strengthen the study. It would be also good to revise some of the results between appendix and main text (i.e., showing the results of using detection threshold choose by training data instead of test data in the main text and vice versa).

---

> ### Author Rebuttal · Authors · 2024-10-24
>
> We thank the reviewer for taking the time to read the paper and for the comments. Below are comments and answers which we hope will be clarifying:
>
> Regarding Tables 1 and 2, and the threshold being optimized on the test set, we agree that this is not practical. The results where the threshold is optimized on the training data was given in the appendix based on page limitations but will now be part of the main text, as it should be. The reason for looking at the optimal threshold on the test data was to investigate the best possible result and hence the full potential of each method. We regard this also as an interesting aspect, but we agree that we should put more focus on showing the results when the optimal threshold is estimated, and we have changed the manuscript accordingly. In particular, we have added results where the thresholds are estimated on data never seen by the prediction model, and where the detectors are evaluated on data neither seen by the prediction model nor the detectors. See Tables 3 and 4 in the revised version. The conclusion is however more or less the same, namely that SPARDACUS is in principle on par with the DOCTOR and MSP methods. However, based on these results, we do not longer write that SPARDACUS is slightly better.
>
> As correctly mentioned, the sensitivity of each method with respect to the choice of threshold can be investigated in Figures 3 and 4. In the revised version we will put more focus on this and acknowledge that the performance of the SPARDACUS method is indeed more sensitive than the DOCTOR and MSP method.

---

### Official Review · Reviewer_eUwr · 2024-10-10
**There are valuable contributions in the paper**

**Confidence:** 4

**Summary:**

The paper suggests a way of measuring probability of misclassification of a machine learning model. The paper suggests to use intermediate layers activations in addition to the output layer activations in this process. It is known that when the model outputs uncertain predictions (maximum softmax probability is small) the model is more likely to misclassify, which is what currently known detectors such as MSP and DOCTOR do. Other layer activations, apart from the final softmax, are used in known methods, such as SafetyCage, but there are limitations of this approach which are solved by proposed paper.
The paper proposes to project the activation statistics for each class into 1D spaces for each layer and class to obtain 1D probability densities using SPARDA method; and then use machine learning or statistical tests to classify between two states: correctly predicted and misclassified. This is in contrast to previous work, SafetyCage, which used out of distribution detection method: the proposed direct training of the final misclassification detector on both correctly and incorrectly classified examples is expected to improve robustness and accuracy of the detector.

**Strengths:**

The work contains novel ideas, extending previously known misclassification detectors.
Particularly, a hypothesis test based on density ratio between two distributions (positive and negative) is used, instead of out of distribution test on a single distribution (positive). This additionally leads to having less assumptions, particularly the distributions no longer need to be assumed to be Gaussian.
The ideas are mathematically sound and well described and the experiments further confirm the theoretical results.

**Weaknesses:**

First, there is a fundamental question about the extent to which we can use misclassification detectors to further enhance the prediction quality of the model. If the misclassification detector uses only information from the model itself, in case of misclassification that information will be incorrect and therefore may not be suitable for further judgement.  However, the existance of other misclassification detectors shows that it is indeed possible to get extra information from the model itself. Still it will be nice to touch this question in the paper and provide additional justification about why the proposed detector is able to gather that extra information from the model.

Second, there is a serious problem with tuning threshold on the test set, as described in line 308 mentioning "optimal threshold on the test data". The paper also mentions " threshold optimized on the training data" in line 319. I believe that both variants are not correct: test threshold shall never be used according to the golden rule of not using the testing set for any calculations that affect final predictions (would be better to remove such results); and training threshold is not optimal as suggested, due to possible overfitting.  Instead, validation set should be used for this purpose. On Figures 3,4 we see that proposed method is very sensitive to the threshold, which additionally confirms the fact that threshold should be re-evaluated correctly on correct data subset.

Additionally, evaluation is limited to two specific cases of image datasets: MNIST and CIFAR-10. would be nice to test the method on non-image datasets, such as the openly available datasets from the UCI datasets repository.

**Final Rebuttal Confidence:**

5

**Final Rebuttal Justification:**

- The rebuttal well addressed the main concern of multiple reviewers: threshold selection is now properly done on a separate validation dataset. However, limited datasets reviewer's question was not addressed. Additionally, proper theshold selection revealed that proposed method does not outperform all competitors on the given datasets.
- My final rating is not changing, recommending acceptance. Lack of additional datasets evaluation and revealed updated results of comparison with other methods beg a question of requesting a further clarification on the part of the experiments, therefore I would highly encourage authors to additionally address the question of limited datasets evaluation (evaluate at least on another non-image dataset). This however does not invalidate the theoretical contribution, justifying the currently given rating.

**Justification:**

The results are novel enough: improving both classical misclassification detectors, such as MSP, as well as modern detectors such as SafetyCage.
There are problems related to threshold selection, mentioned in the Weaknesses section, but I believe that they can be easily corrected in the final version of the paper.
Evaluation on additional non-image datasets would additionally raise credibility of the proposed method, but is enough to justify proposed theoretical contributions.

---

> ### Author Rebuttal · Authors · 2024-10-24
>
> We thank the reviewer for taking the time to read the paper and for the comments. Below are comments and answers which we hope will be clarifying:
>
> Regarding the fundamental question whether we can use misclassification detection to further enhance the ML model. The way we see it is that a trained ML model can be good at capturing some important features, however for some reason it is not as good as capturing other important features. What we would like ideally is to exactly learn these vulnerabilities; not only what they are, but from where in the model they come from. We believe this type of information would be valuable for tuning the model further, not by only retraining with samples we know the ML model is bad at predicting, but also by inspecting the regions in the ML model that actually is the reason for the predictions being wrong. Unfortunately, we were not able to get there in this work, but we regard it as an interesting research direction to observe the patterns within the different layers in the neural network, and not only the output layer, when the predictions are correct and wrong. We have not yet seen research that goes in particular in this direction even though there is plenty of work trying to explain which parts of an image (or layer for that sake) that is important to get a particular prediction (Grad-CAM for instance). However, this is not exactly the same approach even though this can also be used to explore when and where predictions are wrong. We further argue that the SPARDACUS method does not only use information from the model itself as this procedure explicitly use historic information of when the model predicts wrongly. The MSP and DOCTOR methods do not use this information explicitly, only the predictions in the output layer.
>
> Regarding the tuning of the threshold. We agree with the reviewer that overfitting may happen when the threshold is optimized on the training data. We have therefore added results in Table 3 and 4 in the revised version where the threshold values of the detectors are first optimized on data never seen by the prediction model, and the detectors are evaluated on data neither seen by the prediction model nor the detectors. Based on these results, we no longer write that SPARDACUS is slightly better than DOCTOR and MSP, but we regard it as more factual to write that the methods are performing in principle equally well. We still argue that the results for the optimal threshold on the test data is still of certain interest to see the full potential of each method, and it further gave rise to Figures 3 and 4 where we could see the sensitivity in the performance with respect to the threshold value, as the reviewer is correctly pointing out. However these tables are now moved to the appendix.

---

### Official Review · Reviewer_CqCn · 2024-10-15
**Review of SPARDACUS SafetyCage: A new misclassification detector**

**Confidence:** 3

**Summary:**

The authors propose SPARDACUS SafetyCage, a new method for predicting when a trained neural network will perform a missclassification _before it happens_. The method builds on SafetyCage by not only leveraging correctly predicted samples, but also incorrectly predicted samples to estimate two PDFs on which the authors use classical statistical tests to predict if new samples will be correctly or incorrectly predicted.

**Strengths:**

* The empirical results appear to show that the proposed method slightly outperforms previously proposed methods.

* The research direction is interesting and useful for real life scenarios.

**Weaknesses:**

Although the proposed method shows overall improvements, I find that the work currently has a number of overstatements, conflicting statements and is written in a way that makes it difficult to understand the method/process of the method. I elaborate on this here and refer to the bottom of the weaknesses for minor comments.

* One particular conflicting statement which is highly important is that the authors in the abstract write "Importantly, while most existing methods act on the output layer only, our method can be applied on any layer in the neural network, thus being useful in applications, such as feature extraction, that necessarily exploit the intermediate (hidden) layers." but then in the introduction write that SafetyCage works on all layers. Could the authors please clarify which of these are true? Additionally, could the authors comment on why they think this is important when they empirically show that only using the output layer performs the best anyways?
* In general I find the methods section highly difficult to follow, due to the fact that the method is largely described with words rather than mathematics. I think the method section would significantly benefit from including an algorithm figure for example or similar. If I was to implement this method myself, I would find it difficult to do so as the "recipe" is written throughout the text rather than concretely in an algorithm/method figure.
* The authors never clearly describe on which data they estimate the PDFs $f_{i,l}, g_{i,l}$. Could the authors please clarify this?
* The authors have some very strong statements in the conclusion such as:
     * "SPARDACUS has the greatest potential improvement" for which I do not see the reasoning?
     *  " ... it is easy to imagine applications classifying complex inputs (say, sound signals), where an NN is used as a feature-extractor, mapping the input signal to some embedding space, which only later is taken by a specialised model for classification..." but do the authors have any reason to believe this should work based on their hidden layer tests?
     * The authors write "hence the SPARDACUS has greater generalization capabilities", but again I do not see why this should necessarily be the case just because the method can have its projections and PDFs updated. Could the authors comment on this?
     * Also, even though SPARDACUS improves on previously proposed methods I find the final sentence quite overstated "SPARDACUS is in sum a very powerful, extremely flexible state-of-the-art approach" considering the extent of the empirical experiments. I am not saying the empirical results are not significant, but the generalization and proof of working on other modalities, datasets, model sizes etc. is still lacking (which is fine for a first version of a paper, just not for such strong statements in my opinion).
     * Line 398-399 the authors say "but it is clear that this need not be the case for all applications", but again this is not really clear to me, especially based on the empirical results.
* In general I would also say that it is highly preferred that the authors ran multiple seeds when training the models and evaluating the methods in order to get some standard errors on the means presented in Table 1. Currently the methods are very comparable, and it is hard to say if these methods are statistically significantly different (at least subsets of them) - this is especially highlighted by the fact that the results from the original SafetyCage paper are quite different from those reported in this paper. I believe the authors mention that they have the same experimental setup as in the SafetyCage paper, yet the metrics are different, which could indicate that there are somewhat large variations in method performance (although I could have misunderstood this).

Minor comments
* The sentence starting with "Nonetheless.." in 058-061 is unclear to me what the authors are trying to say.
* The authors write mixture Gaussian distribution in 183, but I believe this should be "Gaussian mixture distribution" or "mixture of Gaussians distribution".

**Justification:**

Although the proposed method is novel, there are two key points that are the reason for my score. Firstly, are the previously mentioned overstatements and conflicting points in the paper, which could be addressed relatively easily. More importantly are the considerations on statistical significance of the results which in my opinion only could be addressed by running additional seeds to get standard errors on the different methods. In my opinion this is not an overly large ask considering the scale of these experiments. Should these two main issues be addressed, I would be willing to raise my score.

---

> ### Author Rebuttal · Authors · 2024-10-24
>
> We thank the reviewer for taking the time to read the paper and for the comments. Below are comments and answers which we hope will be clarifying:
>
> Regarding the comment about whether hidden or all layers can be used for the SPARDACUS method. We are not completely sure what aspect the reviewer is referring to, but we acknowledge that one aspect can be seen as conflicting statements the way it was written. In particular, we now explicitly say that SPARDACUS can be used on all hidden layers as well as output layer (see revised abstract). Our results did not include the application of SPARDACUS on the input layer (raw data). Theoretically, we could in principle apply the SPARDACUS framework on the input layer, and we would not get any programmatic errors per se. However, applying SPARDACUS on the input layer is not reasonable for complex input samples such as images. It makes more sense to apply this in intermediate layers where information is compressed.
>
> Regarding why we wanted to look at intermediate layers, we were curious about the degree to which information in the intermediate layers could inform us whether a prediction is correct or not. We can imagine an intermediate layer to give some representation of the data which is important for the ML task such as classification. For instance, one layer could be important to detect a round object in an image. If the detection of a round object is essential for the classification task, it is important that the intermediate layer yields good representations of what is a round object and what is not. If this representation was not good, then one could hypothesize that this layer was important with respect to a prediction being correct or not. We found out later that indeed for our use cases the output layer was the most important. At least in our view, we regard this as an interesting result although it makes sense. However, we still regard the following question as very interesting: Is the output layer always the only layer one should apply for evaluating whether a prediction is correct or not, or could the detection be improved if one also included intermediate layers? We do not think this question can be answered based on our empirical results in this paper. More data sets, models and further inspection and improvement of SPARDACUS would be necessary.
>
> We have followed the reviewer’s suggestion to include algorithms for the SPARDACUS method with the purpose of making it more clear how to construct the method in practice. Specifically, in Algorithm 1 we show how the SPARDACUS method is trained given a data set of samples with known class labels and model predictions. Subsequently, we show how the SPARDACUS method is used to detect false predictions in Algorithm 2. The algorithms are placed in the appendix due to the page limit restriction. We hope these algorithms also makes it easier to understand how the PDFs are generated.
>
> Regarding the statements in the conclusion, we agree with the reviewer that they are too strong and not evidence-based and should be more factual. We have toned down the text, and instead rather emphasize what we regard as more factual, namely that SPARDACUS is more flexible than the MSP and DOCTOR method in terms of having more options to play with. We also instead put more focus on what we regard as interesting future research. As written above, we do think it is interesting whether one can take more advantage of the information in the hidden layers. We have removed the sentence “SPARDACUS is in sum a very powerful, extremely flexible state-of-the-art approach” as we agree with the reviewer that this is an overstatement. The statement referred to in Line 398-399 is also removed. We have generated new results (see tables 3 and 4 in revised version) based on the comments from reviewer eUwr which makes it more factual to say that SPARDACUS is on-par with MSP and DOCTOR, and not slightly better as we phrased it in the first draft.
>
> Regarding the reviewers’ comments about setting multiple seeds on the models based on what the reviewer understands as a difference in the results between the original SafetyCage paper (Johnsen and Remonato, 2023) and the replicated results in this work using the same trained models and datasets. Firstly, we do not see a large difference between the results when comparing the performance metrics for the Mahalanobis method in Johnsen and Remonato and in this work. However, we understand that it may appear as if this is the case because the original work did not apply the output layer in their analysis, only “all”, “hidden” or “penultimate”, while in our work we only presented the best result for each method. It turned out that the Mahalanobis method achieved the best results when only using the output layer. Hence, the results in Johnsen and Remonato using the layers “all”, “hidden” and “penultimate” are not available in our results. When we run the Mahalanobis method using the layers “all”, “hidden” and “penultimate” we indeed get similar results. We have made this fact explicit in the revised manuscript (see lines 313-319 in revised version).
>  Secondly, we assume the reviewer is suggesting to train the prediction model itself several times with multiple seeds. However, we regard it as irrelevant as to how the performance varies in the prediction model itself with different seeds, or for that matter due to other reasons. The model is what is during deployment. What we regard as relevant is the performance in the misclassification detection methods for the same prediction model. The reviewer is correct that the methods MSP, DOCTOR and SPARDACUS are comparable in performance. We make this fact more clear in the revised version.
>
> Regarding the sentence in lines 058-061 starting with “Nonetheless …”, we try to explain an important result discussed in the paper by D. Hendrycks and K. Gimpel that we refer to. This applies to classification models with softmax output layers in which the output vector of the model is the estimated probability for a sample being member of each class. Importantly, they discovered that the maximum softmax value for a particular model output, equal to the class prediction, tended to be larger (statistically) for correct predictions than for wrong predictions. This means in practice that if we plot the (empirical) PDF of the observed maximum softmax values separately for the correct predictions and the wrong predictions (giving two PDFs), we will typically see that the difference in mean between the PDFs are not zero, but that the mean of the PDF of correctly classified predictions is larger than the mean for the PDF for the wrong predictions. This is indeed intuitive, but this is nonetheless the basic idea behind the Maximum Softmax Probability (MSP) Detector: Simply set a threshold and declare a prediction as incorrect whenever the softmax value is less than this threshold. The task is then to set the threshold that maximizes the correctness of the detector. What may be surprising though is what the optimal threshold can be. In fact, it may be very large and close to 1, showing the “overconfidence” of the prediction model. The authors and several others argue that the reason is due to properties in the softmax function.
>
> We thank the reviewer for making us aware that “mixture Gaussian distribution” is the wrong name for what is usually called Gaussian mixture model (GMM). We have corrected this and will now refer to this PDF as a Gaussian mixture model.

---

### Meta-Review · Area_Chair_4eJ2 · 2024-10-28

**Recommendation:** Accept (Poster)
**Confidence:** 3

**Metareview:**

This paper addresses the problem of reliability and misclassification detection in machine learning. The reviewers appreciate the novelty, the writing, and the research direction. After the review and rebuttal stage, the reviewers reached a consensus to accept the paper. The reviewers have some final questions and concerns that could be addressed. For example, one novelty claim is that the proposed method can be used in all layer, but this also seems the case for SafetyCage, hence some clarification is required. There are also shared concerns about hyperparameter tuning. The AC agrees with the positive assessment and encourages the authors to update the paper based on the pointers by the reviewers.

**Suggested Changes To The Recommendation:**

3: I agree that the recommendation could be moved up

---

### Decision · Program_Chairs · 2024-11-06

**Decision:**

Accept (Oral)

**Comment:**

Given the AC positive recommendation, we recommend an oral and a poster presentation given the AC and reviewers recommendations.